# Potential Candidate Molecule of Photosystem II Inhibitor Herbicide—Brassicanate A Sulfoxide

**DOI:** 10.3390/ijms25042400

**Published:** 2024-02-18

**Authors:** Yu Wang, Dong Wang, Baozhu Dong, Jianxiu Hao, Xinyu Jia, Hongyou Zhou

**Affiliations:** Key Laboratory of Biopesticide Creation and Resource Utilization for Autonomous Region Higher Education Institutions, College of Horticulture and Plant Protection, Inner Mongolia Agricultural University, Hohhot 010018, China

**Keywords:** brassicanate A sulfoxide, herbicide, molecular dynamics simulation, PSBD1, binding energy, baculovirus/insect cells system, biolayer interferometry

## Abstract

Brassicanate A sulfoxide, a secondary metabolite of broccoli, exhibited the inhibition of weed growth, but its mechanism of action on weeds remains unclear. To elucidate the mechanism by which brassicanate A sulfoxide suppresses weeds, this study explores the interaction between brassicanate A sulfoxide and the photosystem II D1 protein through molecular docking and molecular dynamics simulations. This research demonstrates that brassicanate A sulfoxide interacts with the photosystem II D1 protein by forming hydrogen bonds with Phe-261 and His-214. The successful expression of the photosystem II D1 protein in an insect cell/baculovirus system validated the molecular docking and dynamics simulations. Biolayer interferometry experiments elucidated that the affinity constant of brassicanate A sulfoxide with photosystem II was 2.69 × 10^−3^ M, suggesting that brassicanate A sulfoxide can stably bind to the photosystem II D1 protein. The findings of this study contribute to the understanding of the mode of action of brassicanate A sulfoxide and also aid in the development of natural-product-based photosynthesis-inhibiting herbicides.

## 1. Introduction

Chemical herbicides have played a crucial role in mitigating the threat of weeds, fostering the flourishing development of agriculture [1]. However, while these chemical herbicides bring benefits to humans, they also have adverse effects on the human living environment [2,3]. Therefore, the development and utilization of environmentally friendly herbicides were the main focus for advancements in herbicides.

Given the significant challenges posed by existing synthetic herbicides, the ongoing search for natural herbicides as alternatives to chemical herbicides is deemed essential for long-term weed control strategies. Allelochemicals are secondary metabolites produced by different parts of plants through secretion, volatilization, and leaching, and they may affect the development of nearby plants [3,4,5]. Our research team has isolated a secondary metabolite from broccoli called brassicanate A sulfoxide, which is a sulfur-containing indole alkaloid with strong phytotoxicity against weeds and model plants [6]; however, the binding site of brassicanate A sulfoxide in plants still remains unclear.

Photosystem II inhibitors reversibly bind to the hydrophobic protein of photosystem II [7]. One inhibitor molecule binds to one D1 protein, reversibly competing with the original plastoquinone binding. Herbicides with different chemical structures, or different substituents on the same herbicide, interact with different amino acids [3]. Amino acid residues participate in binding by forming hydrogen bonds or charge-transfer interactions with the substituents of the herbicides [8]. Urea/triazine inhibitors bind near Ser-264 of the D1 protein [9,10], while phenol inhibitors bind near His-215 of the D1 protein [11].

Biolayer interferometry (BLI) is an advanced, label-free, and real-time monitoring technique that facilitates the high-throughput determination of kinetic parameters during molecular interactions [12]. In BLI technology, ligands are immobilized on the surface of a biosensor, while analytes are dissolved in a solution [13]. When ligands bind or dissociate from analytes, there is a change in the thickness or density of the biolayer at the end of the biosensor. The instrument detects these changes to analyze the corresponding kinetic parameters [14]. This method can be used to validate the results of molecular docking and simulations with more convincing evidence.

In this study, we calculate the binding mode of brassicanate A sulfoxide with PSBD1 through computer simulation and plan to express and purify the PSBD1 protein through a baculovirus/insect cell system to validate the affinity between brassicanate A sulfoxide and PSBD1. We aim to provide a theoretical basis for the development of efficient and environmentally friendly herbicides by researching the binding characteristics of brassicanate A sulfoxide.

## 2. Results

### 2.1. The Docking of Brassicanate A Sulfoxide with PSBD1

The core structure of brassicanate A sulfoxide consists of a fused bicyclic system with a six-membered and a five-membered ring. A nitrogen atom is embedded within the five-membered ring, making it a heterocycle. The presence of a carbonyl group adjacent to the nitrogen indicates the functionality of an amide within the ring system. The molecule contains a sulfoxide functional group, which is characterized by a sulfur atom double bonded to an oxygen atom (Figure 1A). Brassicanate A sulfoxide was bound to PSBD1 through hydrogen bonds, forming hydrogen bonds with Phe-261 and His-214. The binding energy calculated by AutoDock was −7.7 kcal/mol, indicating that brassicanate A sulfoxide can form a stable conformation with PSBD1 through hydrogen bonding (Figure 1B).

### 2.2. Stability of the Simulated System

To study the stable binding mode of brassicanate A sulfoxide and PSBD1, dynamic simulations were conducted at various time points, as illustrated in the snapshots (Appendix A). The analysis showed that brassicanate A sulfoxide remained bound to PSBD1 throughout the 50 ns simulation period with no separation occurring. Moreover, there were no drastic changes in the binding site of brassicanate A sulfoxide. In this study, we calculated the root mean square deviation (RMSD) values for the system, which indicates the system’s stability during simulation and whether it reached equilibrium. The RMSD trend showed that at 30 ns, PSBD1’s RMSD value stabilized at around 4.0 Å until the end of the simulation, suggesting that it reached equilibrium after 30 ns with overall RMSD values lower than 2 Å, thus indicating that there were no significant structural changes. The red curve represents the changes in the binding site of brassicanate A sulfoxide relative to its initial position, showing some fluctuations, thereby implying that there were minor adjustments in the brassicanate A sulfoxide binding site, although the overall value was below 1 Å, indicating that there were minor differences in the binding site (Figure 2A). The results for the radius of gyration (Rg) of the amino acids showed an increase in compactness after brassicanate A sulfoxide binding, with a decreasing Rg value that stabilized at 50 ns (Figure 2B). The trend in the solvent-accessible surface area (SASA) indicated a decrease during the simulation, starting from an initial SASA value of 22,000 Å^2^ and reaching 21,750 Å^2^ upon equilibrium under the binding of brassicanate A sulfoxide (Figure 2C). Furthermore, this study analyzed the root mean square fluctuation (RMSF) trends of each amino acid backbone in the protein. It was observed that the RMSF values of amino acid residues Trp-253, Phe-261, and Phe-270, which bind with brassicanate A sulfoxide, were relatively small, reflecting the stable binding that occurs in that region (Figure 2D).

### 2.3. Binding Mode of Brassicanate A Sulfoxide with PSBD1

When the dynamics simulation stabilized, brassicanate A sulfoxide was primarily bound to the hydrophobic groove of PSBD1 (Figure 3A) due to the charge characteristics of brassicanate A sulfoxide itself. Brassicanate A sulfoxide was bound to the region where the electropositive and electronegative areas of PSBD1 intersect (Figure 3B). When the simulation reached an equilibrium, the specific interactions between brassicanate A sulfoxide and PSBD1 were analyzed. The analysis showed that hydrogen bonding, hydrophobic interactions, and van der Waals contacts occurred between brassicanate A sulfoxide and PSBD1. Specifically, the tertiary amine of brassicanate A sulfoxide undergoes an H-π interaction with the charge center of the side-chain phenyl ring of tryptophan, Trp-253, at a distance of 2.8 Å. The carbonyl group of brassicanate A sulfoxide forms a 1.9 Å hydrogen bond with the backbone amide bond of Phe-261. Additionally, brassicanate A sulfoxide engages in hydrophobic interactions with the surrounding hydrophobic amino acids such as Leu-210, Phe-257, Ile-259, Ala-260, and Phe-270 (Figure 3C,D).

### 2.4. Types and Quantities of Amino Acids Involved in the Interaction between Brassicanate A Sulfoxide and PSBD1

To explore the types and quantities of amino acids involved in the interaction between brassicanate A sulfoxide and PSBD1, kinetic simulations were conducted. These simulations revealed that hydrogen bonding and hydrophobic interactions are predominant between brassicanate A sulfoxide and PSBD1. Throughout the 50 ns simulation, certain amino acids consistently interacted with brassicanate A sulfoxide. For instance, Leu-210, Trp-253, and Phe-270 were continuously present, indicating persistent contact. These amino acids are hydrophobic and do not participate in hydrogen bonding, suggesting that they engage in sustained hydrophobic interactions with brassicanate A sulfoxide. The continuous orange segments which are associated with Phe-261 indicate interactions beyond hydrogen bonding and that are likely hydrophobic in nature. Ile-259 and Ala-260 also showed frequent interactions, highlighting their significance in this process (Figure 4).

### 2.5. Binding Energy of Brassicanate A Sulfoxide with PSBD1

The analysis showed that the binding free energy (ΔG _bind_) between brassicanate A sulfoxide and PSBD1 was −10.43 kcal/mol, indicating a moderately strong binding affinity. In terms of energy components, the binding was mainly facilitated by van der Waals forces (including hydrogen bonding) and electrostatic interactions (π-stacking interactions). The van der Waals interaction ΔE _vdw_ was calculated to be −8.86 kcal/mol, and the electrostatic interaction ΔE _ele_ was calculated to be −4.17 kcal/mol. The positive solvation free energy ΔG _solv_ of 2.61 kcal/mol, which was further divided into polar solvation free energy ΔE _GB_ and non-polar solvation free energy ΔE _surf_, indicated that solvation was not favorable for binding (Table 1).

### 2.6. Expression and Purification of PSBD1

Based on the recombinant information and vector details, the recombinant plasmid of *psbA* should consist of 5238 base pairs. In the competent DH10Bac cells, *psbA* was successfully recombined with the donor plasmid (Figure 5A). Sf9 cells were harvested for cell lysis after sufficient culturing post 96 h of transfection, followed by Western blot and SDS-PAGE analysis. The Western blot results from P2 progeny cells containing *psbA* (Figure 5B,C) revealed bands with His tags that correspond in size to the construct information. This indicates the successful transfection of plasmids into cells. After purification using a Ni column, 0.02 mg/mL of PSBD1 protein was obtained, which is suitable for subsequent validation experiments.

### 2.7. Validation of the Binding between PSBD1 and Brassicanate A Sulfoxide

To verify the accuracy of the molecular docking and simulation results of brassicanate A sulfoxide with PSBD1, we tested the affinity of brassicanate A sulfoxide with PSBD1. The results showed that the affinity constant between brassicanate A sulfoxide and PSBD1 was 2.69 × 10^−3^ M, proving that there was a binding phenomenon between brassicanate A sulfoxide and PSBD1, even though the dissociation was relatively slow (Figure 6A). (-)-Spirobrassinin, due to its similar structure to brassicanate A sulfoxide, could also bind with PSBD1 (Figure 6B).

## 3. Discussion

The compound brassicanate A sulfoxide is a unique secondary metabolite found exclusively in cruciferous plants and was first isolated from radishes [15]. Our research team has demonstrated its ability to inhibit weed growth [6]. To further investigate the interaction between brassicanate A sulfoxide and PSBD1, we conducted more in-depth experimental and simulation studies. Initially, our molecular dynamics simulations revealed the dynamic behavior of brassicanate A sulfoxide and PSBD1 and how their interactions with key amino acid residues evolve over time.

The indole group and sulfoxide group in brassicanate A sulfoxide are key functional groups that generate herbicidal activity. The indole group in brassicanate A sulfoxide interacted with the surrounding amino acid residues through hydrophobic forces, thereby binding to the active pocket of PSBD1. This was similarly demonstrated in the molecular docking of indole nitrogen compounds with PSBD1 [16]. Interestingly, there are currently several commercial herbicides containing sulfoxide groups, such as pyrasulfotole, lancotrione, topramezone, and benzobicyclon [17], as well as aryl sulfoxides that exhibit excellent herbicidal activity [18]. Molecular dynamics simulations have shown that the oxygen atom of the sulfoxide group forms a hydrogen bond with a hydrogen atom in Phe-261. (-)-Spirobrassinin, another secondary metabolite of cruciferous plants with a structure similar to brassicanate A sulfoxide, engages in π-π interactions with the phenyl ring of Phe-270 during its interaction with PSBD1 [19]. The sulfoxide group, being an electron-donating group, plays a key role in herbicidal activity by forming hydrogen bonds with Phe-261 in the active pocket of PSBD1. The presence of electron-donating groups such as nitro or amide groups can alter certain physical and chemical properties such as molecular size, hydrogen bonding, ionization state, hydrophobicity, and water solubility to enhance herbicidal activity [20,21,22]

The molecular dynamics simulation of brassicanate A sulfoxide and PSBD1 suggested that there were unexplored target sites in the D1 protein, such as Phe-261 and Thr-253, which have a strong binding affinity with PSBD1 (Figure 5). However, brassicanate A sulfoxide can also form hydrogen bonds with His-214. Between 20 ns and 35 ns, the interaction between brassicanate A sulfoxide and PSBD1 weakens, while the interaction between the former and Phe-261 strengthens, proving that Phe-261 can form hydrogen bonds with both the sulfonyl and carbonyl groups (Figure 6). This could be an effective solution against weeds resistant to PSII inhibitors. The hydrogen bonds formed by brassicanate A sulfoxide might inhibit the function of the D1 protein. A docking analysis showed that brassicanate A sulfoxide formed two hydrogen bonds with PSBD1, which may have played a role in inhibiting its photosynthetic activity, ultimately suppressing weed growth. Similar to the amino acids (Ser-264 and His-214) targeted by PSII inhibitor herbicides, resistance in weeds has been reported [23,24,25]. Thus, the action sites provided by brassicanate A sulfoxide could be used to develop natural derivatives for weed control.

To verify the binding of brassicanate A sulfoxide with PSBD1, we have for the first time expressed PSBD1 from *Thermophilic cyanobacteria* in insect cells. Insect cells were capable of complex post-transcriptional modifications, including the correct folding, glycosylation, and the formation of disulfide bonds, all of which are crucial for the functional integrity of many membrane proteins [26]. Compared to bacterial expression systems, these systems generally yield higher amounts of membrane proteins, which was particularly important for proteins that are difficult to express in large quantities [27]. Since insect cells are eukaryotic, they are more suitable for expressing eukaryotic membrane proteins; they can correctly process and modify these proteins. Additionally, insect cell cultures can be relatively easily scaled up, allowing for the large-scale production of membrane proteins, which was beneficial for studying drug targets [28]. However, it is crucial to note that the PSBD1 expressed in insect cells and in *Thermophilic cyanobacteria* are subjected to different physiological conditions, and whether the expressed PSBD1 retains its physiological function is uncertain. Therefore, in subsequent studies, we need to validate the functionality of PSBD1. This can be achieved by biochemical assays to measure the activity of PSBD1 or through molecular biology methods to assess the gene expression levels of PSBD1. The successful expression of PSBD1 in insect cells provides strong support for our in-depth investigation into the mechanism of action of brassicanate A sulfoxide.

In summary, this study will provide new insights into the mechanism of action of brassicanate A sulfoxide, while also serving as a reference for other scientists. As PSBD1 alone cannot perform its corresponding function in vitro, we used biolayer interferometry technology to analyze the affinity between brassicanate A sulfoxide and PSBD1, validating the results of the molecular docking and simulation calculations. The binding between brassicanate A sulfoxide and PSBD1 is of moderate strength, demonstrating that brassicanate A sulfoxide can act on PSBD1. Through these studies and experimental evidence, we can demonstrate that brassicanate A sulfoxide can form hydrogen bonds with key amino acid residues in PSBD1, potentially making it a candidate molecule for photosynthesis inhibitors.

## 4. Materials and Methods

### 4.1. Molecular Docking

The structure files of PSBD1 (PDB code 7ZCL) were obtained from the RCSB Protein Data Bank (https://www.rcsb.org/) accessed on 29 July 2023. Additionally, the structure of brassicanate A sulfoxide was downloaded in sdf format from the PubChem database (https://pubchem.ncbi.nlm.nih.gov/) accessed on 30 July 2023. For the docking simulations, AutoDock was used to append necessary hydrogen atoms to the structures. The grid maps for the binding affinity calculations were created using Autogrid, with the grid dimensions set to 58 × 60 × 62 Å and a grid point spacing of 0.375 Å. The van der Waals and electrostatic interactions were computed using the AutoDock parameter set and a distance-dependent dielectric function, respectively [29]. The docking simulations were performed using the Lamarckian genetic algorithm and the Solis and Wets local search method [30].

### 4.2. Preparation of PSBD1 and Brassicanate A Sulfoxide Structure

UCSF Chimera-1.15 software was utilized to edit PSBD1’s structure by removing extraneous atoms, thus preserving only the relevant structure. Atomic charges for PSBD1 were then calculated using the AMBER14SB force field. To determine the pKa values of amino acids at a neutral pH of 7.0, the H++3 online tool was employed [31]. For brassicanate A sulfoxide, its three-dimensional structure was created using RDKit 2019.03, an open source cheminformatics tool. This structure underwent conformational sampling, followed by optimization using the MMFF94 force field, which resulted in low-energy conformers [32]. Finally, UCSF Chimera was again used to assign AM1-BCC partial charges to these structures [33].

### 4.3. Molecular Dynamics Simulations

To investigate the stable interaction between PSBD1 and brassicanate A sulfoxide, molecular dynamics simulations were conducted using Gromacs 5.1.5 [34], an open source software. The simulation setup involved a periodic box in a closed environment with the following conditions set: a temperature of 289.15 K, pH of 7.0, and atmospheric pressure of 1 bar. PSBD1 was placed at the center of the box, ensuring a minimum distance of 0.1 nm from the box edges. The structure of the receptor (PSBD1) was processed into a GROMACS-compatible format using pdb2gmx while applying the AMBERff14SB [35] force field parameters. Brassicanate A sulfoxide was converted into itp-format topology files that are recognizable by GROMACS using AmberTools [36], while the ligand atoms were processed using the GAFF force field [37]. Initially, the system’s energy was minimized using the steepest descent method. This was followed by constrained molecular dynamics simulations for 1000 ps in the NVT ensemble (constant number of particles, volume, and temperature), with protein positions being restrained. Subsequently, another 1000 ps simulation was run in the NPT ensemble (constant number of particles, pressure, and temperature). After these equilibration phases, both the wild-type and mutant systems underwent 50 ns of production molecular dynamics simulations with a 2 fs time step. The linear constraint solver algorithm was employed for constraining the covalent bond lengths, while long-range electrostatic interactions were managed using the Particle Mesh Ewald method [38]. For the analysis, contact atom details were examined using VMD 1.9.3, and interaction types were analyzed with PyMOL 2.04. Post-simulation, the gmx module was used to compute various parameters, including the radius of gyration, hydrophobic contacts, root mean square deviation, and root mean square fluctuation.

### 4.4. Binding Free Energy Calculation

The calculation of the binding free energy was carried out using Molecular Mechanics with Generalized Born and Surface Area (MM/GBSA), and the MMPBSA.py program was employed to generate the following formula [39,40]. The formula used was ∆G _bind_ = ∆H − T∆S ≈ ∆G _solv_ + ∆G _GAS_ − T∆S; ∆G _GAS_ = ∆E _int_ + ΔE _vdw_ + ΔE _ele_; and ΔG _solv_ = ∆E _surf_ +∆E _GB_. Here, ∆G _GAS_ signifies the change in gas-phase dynamic energy, broken down into internal energy change (∆E _int_), van der Waals energy variation (∆E _vdw_), and changes in electrostatic energy (∆E _ele_). ∆G_solv_ accounts for the influence of the solvent, split into the polar component (∆E _GB_) and the nonpolar component (∆E _surf_). The calculation of ΔE _GB_ was performed separately using the APBS program for that section. In contrast, ∆E _surf_ is determined by measuring the solvent-accessible surface area. This approach provides a comprehensive view of the energetic changes associated with the binding process.

### 4.5. The Expression of PSBD1

PSBD1 was expressed using a baculovirus/insect cells system. pFastBac^TM^1 was digested with BamHI and HindIII, and then *psbA* was cloned into the corresponding sites. The constructed expression vector was transformed into DH10Bac competent cells and shaken at 37 °C at 225 rpm for 4 h. Then, 100 µL of the culture was spread on LB plates containing 50 µg/mL kanamycin, 7 µg/mL gentamicin, 10 µg/mL tetracycline, 24 µg/mL IPTG, and 20 µg/mL X-Gal, and incubated while inverted at 37 °C for 48 h. White single colonies were picked from the plates and inoculated into LB medium containing 50 µg/mL kanamycin, 7 µg/mL gentamicin, and 10 µg/mL tetracycline, and shaken at 37 °C at 225 rpm for 4 h [41]. Single colonies were picked for the M13 F and M13 R identification of the recombinant plasmid (Appendix A).

### 4.6. Purification of PSBD1

0.5 × 10^6^/mL Sf9 cells, in the logarithmic growth phase with a cell viability greater than 95%, was inoculated into a six-well cell culture plate and incubated at 27 °C for 1 h to allow the cells to adhere. A complex of Bacmid and Cellfectin@IReagent was prepared and added to each well. This was incubated at 27 °C until the cells were infected with the virus, then the supernatant collected to be used as the P1 virus. 20 mL of Sf9 cells was inoculated at a density of 2.0 × 10^6^/mL into a 250 mL flask, an appropriate amount of P1 virus was added to amplify the P2 virus, and it was incubated at 27 °C and shaken at 120 rpm for 4 d. The culture was collected by centrifugation, the virus-containing supernatant was transferred to another sterile 50 mL tube, and the P2 virus supernatant was stored at 4 °C and protected from light. The pellet was resuspended in buffer (PBS, pH 6.0), disrupted by sonication, then centrifuged and checked for expression using SDS-PAGE and Western blot. 200 mL of Sf9 cells was inoculated at a density of 2.0 × 10^6^/mL into a 1 L flask, and an appropriate amount of P2 virus was added to amplify the expression. This was incubated at 27 °C and shaken at 120 rpm for 4 d. After centrifugation, the supernatant and pellet were collected separately. The pellet was resuspended in buffer (PBS, pH 6.0), disrupted by sonication, and then centrifuged. The supernatant was purified using a Ni column, and the purification was checked using SDS-PAGE and Western blot.

### 4.7. Biolayer Interferometry

Using Octet R8 (Sartorius BioAnalytical Instruments Inc), the affinity between PSBD1 and brassicanate A sulfoxide and (-)-Spirobrassinin was analyzed. Initially, the PSBD1 protein was diluted to 20 µg/mL and immobilized in a 96-well black, non-transparent plate for 600 s. Brassicanate A sulfoxide was diluted to 1 mM in PBS with a 5% DMSO buffer. The control wells received PBS with 5% DMSO, while the sample wells received the diluted brassicanate A sulfoxide. Due to the presence of a His tag on PSBD1, the specific capture of the PSBD1 protein was achieved using an NTA sensor. After reaching a signal of 5 nm, binding with brassicanate A sulfoxide occurred. Sensor regeneration was achieved through the process of cleaning the sensor with nickel chloride and glycine·HCl. In this experiment, a total of 6 sensors were used, and the sensors were wetted 10 min before the experiment using a wetting buffer consisting of 1×PBS.

## 5. Conclusions

In this study, we demonstrated the stable binding of brassicanate A sulfoxide with PSBD1. Brassicanate A sulfoxide forms hydrogen bonds with His-214 and Phe-261 in the active cavity of PSBD1 through the oxygen atom of its sulfoxide group. Notably, His-214, also a key amino acid in the activity of plastoquinone, serves as a crucial target for herbicides. Additionally, we successfully expressed PSBD1 exogenously, and brassicanate A sulfoxide exhibited a strong affinity for PSBD1. Our experiments clearly indicate that brassicanate A sulfoxide, as a small molecule, can serve as a promising candidate for a photosystem II inhibitory herbicide (Figure 7), providing a compound basis for the development of natural product herbicides.

## Figures and Tables

**Figure 1 ijms-25-02400-f001:**
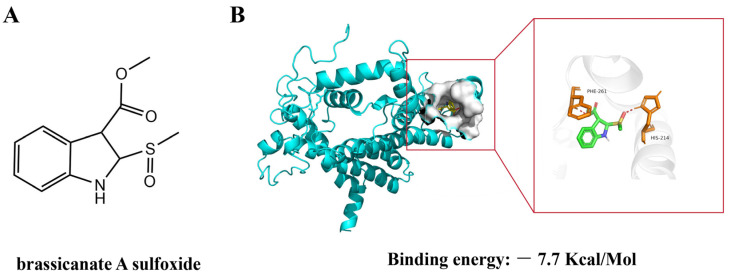
Molecular docking of brassicanate A sulfoxide with PSBD1. (**A**) Chemical structure of brassicanate A sulfoxide; (**B**) binding mode of brassicanate A sulfoxide with PSBD1. The binding energy was predicted based on AutoDock calculations.

**Figure 2 ijms-25-02400-f002:**
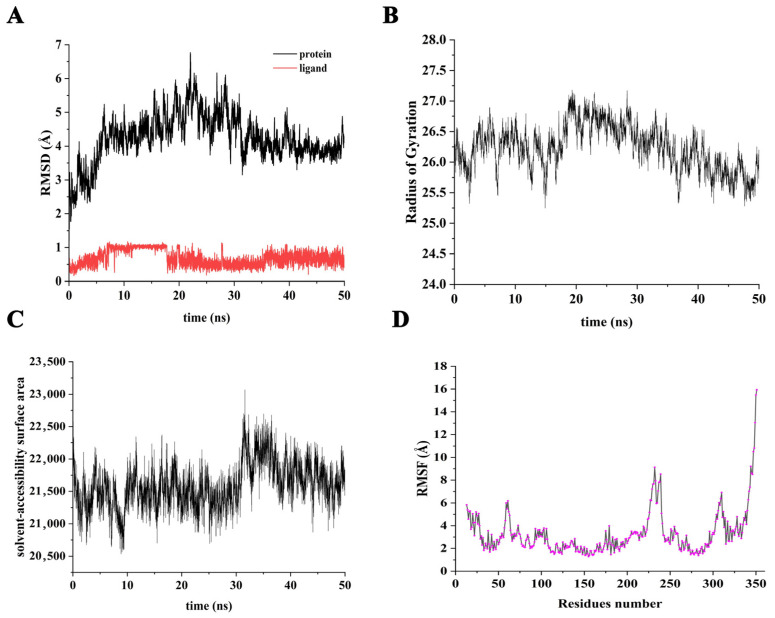
Stability during simulation period. (**A**) The trend of RMSD values for the molecules in the system during the simulation period; (**B**) the trend of PSBD1′s Rg values during the simulation period; (**C**) the trend of PSBD1′s SASA values during the simulation period; (**D**) the statistical analysis of the RMSF of each amino acid in PSBD1 during the simulation period.

**Figure 3 ijms-25-02400-f003:**
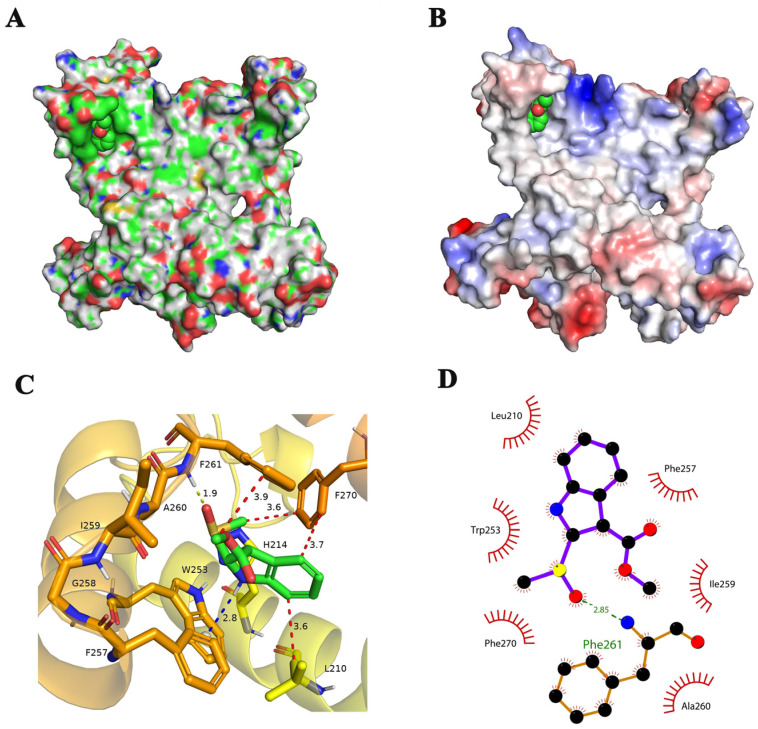
Analysis of the binding mode between brassicanate A sulfoxide and PSBD1 during simulations. (**A**) Distribution of hydrophobic and hydrophilic regions in PSBD1, with green indicating hydrophobic regions and magenta indicating hydrophilic regions; (**B**) electrostatic potential surface of PSBD1, with red indicating negatively charged regions and blue indicating positively charged regions; (**C**) 3D interaction diagram of brassicanate A sulfoxide with PSBD1, where yellow dashed lines indicate hydrogen bonds and numerical values indicate the bond distances; (**D**) 2D interaction diagram, where green dashed lines indicate hydrogen bonds and eyelashes indicate hydrophobic interaction groups.

**Figure 4 ijms-25-02400-f004:**
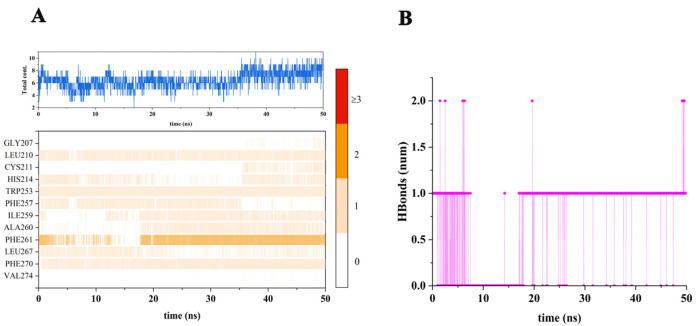
Types and quantities of amino acids involved in the interaction between brassicanate A sulfoxide and PSBD1. (**A**) Statistics of amino acid contacts between brassicanate A sulfoxide and PSBD1 during dynamic simulation; (**B**) statistics of hydrogen bond numbers between brassicanate A sulfoxide and PSBD1.

**Figure 5 ijms-25-02400-f005:**
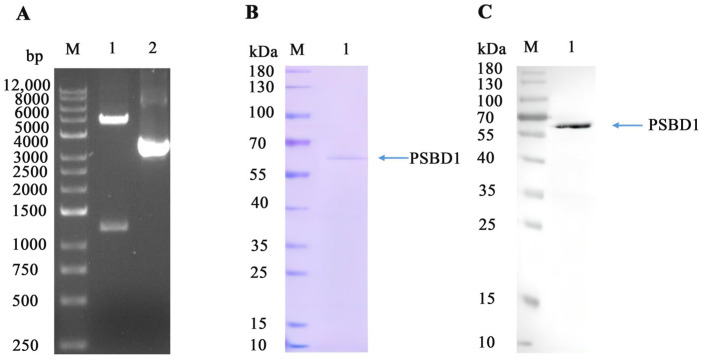
Expression and Purification of PSBD1. (**A**) Gel electrophoresis of recombinant plasmid (M), DL12000 (1), and plasmid (2), digested with BamHI-HindIII; (**B**) SDS-PAGE of PSBD1; (**C**), Western blot of PSBD1.

**Figure 6 ijms-25-02400-f006:**
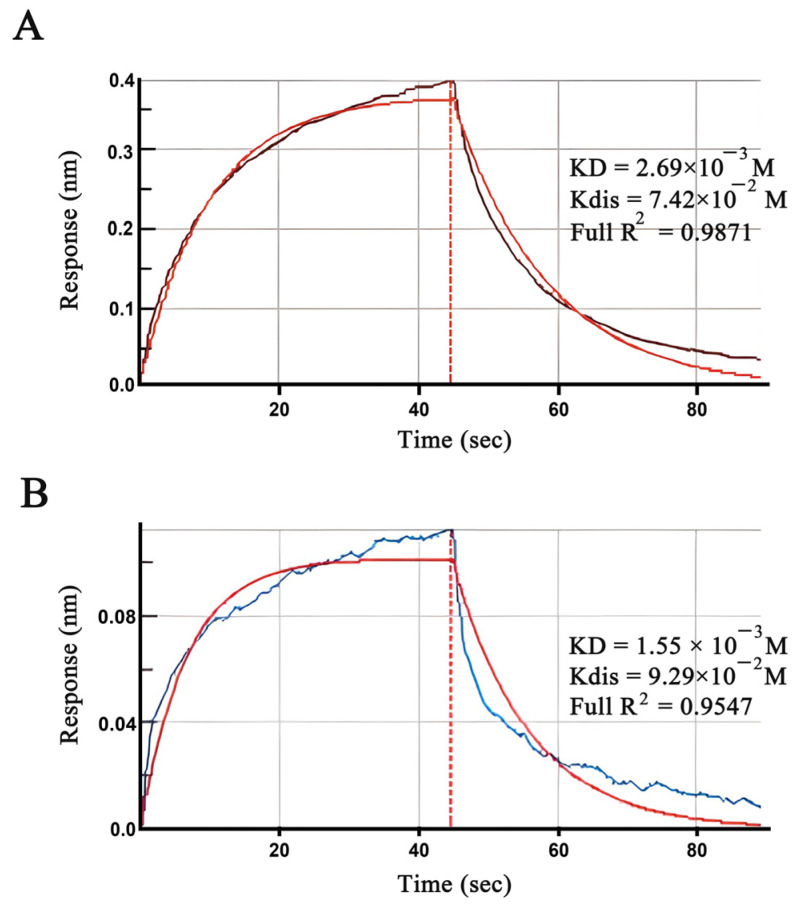
Determination of the binding of (-)-Spirobrassinin and brassicanate A sulfoxide to PSBD1. (**A**) Response values for the binding and dissociation of brassicanate A sulfoxide to PSBD1; (**B**) response values for the binding and dissociation of (-)-Spirobrassinin to PSBD1. Blue indicates the signal of the interaction between the immobilized sensor and (-)-Spirobrassinin, reddish-brown indicates the signal of the interaction between the immobilized sensor and brassicanate A sulfoxide, and red indicates the signal of the interaction between the immobilized sensor and PBS + 5% DMSO.

**Figure 7 ijms-25-02400-f007:**
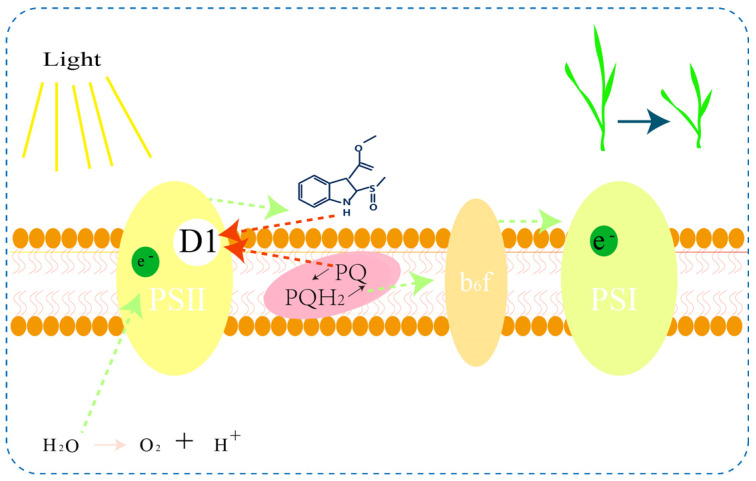
Brassicanate A sulfoxide competes to bind with PSBD1. The green dashed lines indicate the pathway of electron transfer, while the red dashed lines indicate the competitive binding of PQ and brassicanate A sulfoxide with the PSBD1 protein.

**Table 1 ijms-25-02400-t001:** The binding free energy between brassicanate A sulfoxide and PSBD1.

Binding Energy	Value
ΔE _vdw_	−8.86
ΔE _ele_	−4.17
ΔE _GB_	6.63
ΔE _surf_	−4.02
ΔE _gas_	−13.03
ΔG _solv_	2.61
ΔG _bind_	−10.43

Each value is the mean ± standard error. The binding free energies are predictions based on MM/GBSA calculations.

## Data Availability

Data are available within the article or its Appendix A.

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
