# Peer review of "Potential Candidate Molecule of Photosystem II Inhibitor Herbicide—Brassicanate A Sulfoxide"

_ijms, 2024, doi:10.3390/ijms25042400_

Round 1
Reviewer 1 Report
Comments and Suggestions for Authors
The manuscript of Hongyou Zhou et al “Potential Candidate Molecule of Photosystem II Inhibitor Herbicide — Brassicanate A Sulfoxide” have presented a study deals with the study of the mechanism of action of brassicanate A sulfoxide, a secondary metabolite of broccoli, exhibited activity in inhibiting weed growth. The previous results presented by the authors showed that brassicanate A sulfoxide exerted herbicidal activity by affecting the activity of DHAD, HPPD and ALS enzymes. It is well known that herbicide targets also include components of photosystem II (PSBD1), where they compete with plastoquinone for binding sites, disrupt the electron transport chain, and interfere with light reactions.
In this study the binding mode of brassicanate A sulfoxide with PSBD1 was calculated through computer simulation. Hydrogen bonds formation with key amino acid residues in PSBD1 was demonstrated. To investigate the stable interaction between PSBD1 and brassicanate A sulfoxide, molecular dynamics simulations was conducted. Of interest was the use of Bio-Layer Interferometry technology to analyze the affinity between brassicanate A sulfoxide and PSBD1. This experiments elucidated that brassicanate A sulfoxide can stably bind to the photosystem II D1 protein.
In addition, PSBD1 was expressed and purified using a baculovirus/insect cells system to validate the affinity between brassicanate A sulfoxide and PSBD1.
This manuscript is a suitable work for International Journal of Molecular Science.
I noticed just two points in the manuscript, which must be clarified:
1. The Discussion needs some adjustment. For example, p.7. line 207-210. “However, the phenyl ring of brassicanate A sulfoxide does not form any non-covalent bonds with any amino acid residues. While both π-π interactions and hydrogen bonds are non-covalent, the hydrogen bond is formed with the sulfoxide oxygen as a hydrogen bond acceptor is more polar and has stronger binding capabilities.” This sentence, and the discussion of the entire paragraph, should be discussed with the addition of some References, bearing in mind the herbicidal properties and the mechanism of action of compounds of tetrahydroindole and dihydroindole structures and also synthetic herbicides with a sulfoxide function.
2. The References must be carefully checking. Ref [18] –page numbers must be added; Ref [22] – DOI must be added.
Comments on the Quality of English LanguageEnglish language is correct. Minor editing of English language required.
Reviewer 2 Report
Comments and Suggestions for Authors
This manuscript continues the works by the same group of authors - and, actually, they seem to be the only team attempting to explore the potential of this compound (Brassicanate A Sulfoxide) as a herbicide. Previously, the authors published one paper (https://www.mdpi.com/2223-7747/12/13/2576) indicating that among other compounds in Brassica oleracea L. this one can be especially promising - in this manuscript they validate this hypotheses by means of both theoretical and experimental methods.
I do not see any experimental flaws in this work, nor self-plagiarism (this is a really new part of research on this topic). It likely will be of interest for specialists in herbicides and, generally, biologically active compounds. I recommend acceptance in present form.
Comments on the Quality of English Language
minor spell checking required
Reviewer 3 Report
Comments and Suggestions for Authors
The work by Zhou and coworkers provides a well-dessinged computational/experimental study to asses the potential inhibition of PSII by Brassicanate A Sulfoxide. The proposed methods and correctly applied and the results obtained and properly analyzed, leading to robust conclusions. I think the paper should be considered for publication, just after a few minor aspects are addressed:
- The numerical values provided include too many unnecesary significant figures. Having data such as "−10.4258 kcal/mol" includes decimals figures that are meaningless (1-2 decimals figures are more sensible in these cases)
- The computation details on the calculation of interaction energies should be clarified. It seems that the authors used APBS program to compute (some of/all) the terms contributing to the free energy of binding. But they also mention the use of gromacs tools in this section. Were such tools also used to get any of the energetic terms. Please, clarify these details.
